

# INFLUENCE OF THE SEMIDIURNAL LUNAR TIDE ON THE EQUATORIAL PLASMA BUBBLE ZONAL DRIFTS OVER BRAZIL

Igo Paulino[1], Ana Roberta Paulino[2], Amauri F. Medeiros[1], Cristiano M. Wrasse[3], Ricardo Arlen Buriti[1], and Hisao Takahashi[3]

[1]Unidade Acadêmica de Física, Universidade Federal de Campina Grande, Campina Grande, PB, Brazil
[2]Departamento de Física, Universidade Estadual da Paraíba, Campina Grande, PB, Brazil
[3]Divisão de Clima Espacial, Instituto Nacional de Pesquisas Espacias, São José dos Campos, SP, Brazil

**Correspondence:** Igo Paulino (igo.paulino@df.ufcg.edu.br)

**Abstract.** Using OI6300 airglow images collected over São João do Cariri (7.4$^o$S, 36.5$^o$W) from 2000 to 2007, the equatorial plasma bubble (EPB) zonal drifts were calculated. A strong day-to-day variability was observed in the EPB zonal drifts due to the complexity in the dynamics of the nighttime thermosphere-ionosphere system near the equator. The present work investigated the contribution of the semidiurnal lunar tide $M_2$ for the EPB zonal drifts. On average, the $M_2$ contributes 5.6% to the variability of the EPB zonal drifts, presenting an amplitude of 3.1 m/s. The results showed that the $M_2$ amplitudes in the EPB zonal drifts were solar cycle and seasonal dependents. The amplitude of the $M_2$ was stronger during the high solar activity reaching over 10% of the EPB zonal drift average. Regarding the seasons, during the southern hemisphere summer, the $M_2$ amplitude was twice larger (12%) compared to the equinox ones. The seasonality agrees with other observations of the $M_2$ in the ionospheric parameters such as vertical drifts and electron concentration, for instance. On the other hand, the very large $M_2$ amplitudes found during the high solar activity must be further investigated.

## 1 Introduction

Equatorial plasma bubbles (EPBs) appear during the nighttime near the magnetic equator and extend across the tropics along the magnetic field lines (e.g., Weber et al., 1978). They consist in plasma density depletion compared to the background ionosphere (e.g., Sobral et al., 1980). Therefore, airglow emissions from the thermosphere can be used to detect and study the morphology and dynamics of this phenomenon (e.g., Sobral et al., 1980; Mendillo and Baumgardner, 1982; Fagundes et al., 1995; Takahashi et al., 2001). Additionally, as EPBs represent plasma irregularities, radio techniques have mainly been used to investigate them as well (e.g., Woodman and La Hoz, 1976; Abdu et al., 1985, 1998; Fejer et al., 1996; de Paula and Hysell, 2004; Chu et al., 2005).

In the equatorial ionosphere, the zonal electric field controls the vertical movement of the F layer. In general, during the daytime the plasma moves upward, while during the nighttime, the motion is downward. However, after the sunset, can occur the pre-reversal enhancement (PRE), which is a rapid upward motvement of the F region before it reverses and starts a downward



motion (Farley et al., 1986). The PRE has well defined temporal dependencies being more intense during the summer and high solar activity (Fejer et al., 1991).

Besides the PRE, after the sunset, there is a quick recombination in the ionospheric E region (e.g., Bates, 1988), producing
a strong vertical gradient of plasma with high density levels in the F region. This scenario is very favourable to the Rayleigh-Taylor instability (RTI) development, which has been recognised as the main mechanism to generate EPBs (Dungey, 1956; Haerendel et al., 1992). Even so, the RTI theory requires a seeding process in order to increase its growth rate. Although gravity waves, thermospheric wind, post-sunset vortex, large scale waves and magnetic disturbances have been pointed out as possible seedings for EPBs (e.g., Kudeki et al., 2007; Abdu et al., 2009; Abalde et al., 2009; Saito and Maruyama, 2009; Takahashi
et al., 2009; Paulino et al., 2011a; Huang et al., 2013; Tsunoda et al., 2018), this topic continues under scientific investigation (e.g., Fritts et al., 2009, and references therein),

EPBs move zonally eastward under quiet magnetic conditions during the nighttime, reaching high drift values during the evening time (e.g., Pimenta et al., 2003; Paulino et al., 2011b). Clear seasonal and solar activity dependencies were observed as well (e.g., Pimenta et al., 2001; Paulino et al., 2011b). In contrast, magnetic storms can totally disturb the expected dynamics,
inclusive reversing the drifts to the west (e.g., Abdu et al., 2003; Li et al., 2009; Paulino et al., 2010; Santos et al., 2016).

As the PRE (vertical motion) as the zonal drifts presents a strong day-to-day variability (e.g., Liu, 2020; Aswathy and Manju, 2021, and references therein). The understanding of the short period variability represents one of the biggest challenge in atmospheric and space science (e.g., Tsunoda, 2006). Among the features that can produce day-to-day variabilities, the lunar tide appeared as relevant after the work by Stening and Fejer (2001), which simulated the effects of the semidiurnal lunar tide
($M_2$) on the vertical motion of the F region and PRE. According to them, the $M_2$ can either change the local time of PRE or its amplitude. The $M_2$ has a period of 12.43 hours and it is the main lunar tide periodicity. It is primarily produced in the lower levels of the atmosphere due to the gravitational interaction of the Sun-Earth-Moon system and it can propagate upward into the atmosphere-ionosphere (Chapman and Lindzen, 1970).

As a direct response to the EPBs and M2 interaction, changes of ∼14 min on the starting time of the EPBs were observed
over the Brazilian equatorial region (Paulino et al., 2020) along almost one solar cycle. The present article aims to investigate, for the first time, the $M_2$ effects on the EPB zonal drifts derived from airglow images. It used measurement over São João do Cariri (7.4$^o$S, 36.5$^o$W) from 2000 to 2007, covering periods of high and low solar activities.. In addition, the seasonality is going to be studied as well.

## 2  Observations and methods

An all-sky airglow imager was deployed at São João do Cariri in September 2000 to observe the nighttime airglow. The OI6300 emissions were used to study equatorial plasma bubbles. This imager operated up to December 2010 and it was equipped with a CCD camera and filter wheel. The CCD had a resolution of $1024 \times 1024$ pixels binned on-chip down to $512 \times 512$ to enhance the signal-to-noise ratio. The CCD had high linearity (0.05%), high quantum efficiency, low dark noise (5 electrons per pixel per second) and low readout noise (15 electron rms). In addition, the optical system had a fisheye lens and a telecentric system





of lenses, allowing the record of the OI6300 airglow images, with 90 s of time integration. The filter wheel could select five

other emissions, but only OI6300 filter was used in the present work to calculate the EPB zonal drifts. The observations were

made between September 2000 and April 2007, centered at new moon periods, resulting in thirteen nights of data per month.

The collected images were unwarped to the geographic coordinates with a spatial resolution of $512\,km \times 512\,km$ according

to the method described by Garcia et al. (1997). Then, the EPB zonal drifts were calculated using a cross-correlation between

two lines of consecutive images. The lag between the lines was assumed to be the displacement of the structures within the

time interval recorded between the two images. With this information, it was possible to calculate the EPB zonal drifts across

all latitudes coved by the field of view of the unwarped images (see Paulino et al., 2011a, for further details).

The amplitude of the semidiurnal lunar tide oscillation in the EPB zonal drifts was calculated by converting the solar local

time $t$ to the lunar local time $\tau$, i.e., $\tau = t - \nu$, where $\nu$ is the age of the Moon set to be equals $0$ at the New Moon (see Forbes

et al., 2013; Paulino et al., 2017, for further details).

To investigate a possible influence of the solar cycle on the strength of $M_2$ on EPB zonal drifts, the period of observation

was divided as high solar activity (HSA) and low solar activity (LSA). The classification was done using the F10.7 cm solar

flux as a proxy to the solar activity. The nights with solar flux greater than $140 \times 10-22\ Wm^{-2}Hz^{-1}$ were considered as

HSA. In contrast, the nights with solar flux lower than $80 \times 10-22\ Wm^{-2}Hz^{-1}$ we supposed to be LSA. These criteria make

the HSA period from September 2000 to December 2002 and the LSA period from January to April 2007. Additionally, the

summer period includes the months from December to February. The equinox period was set for March, April, May, September,

October and November for the whole period of observations.

## 3  Results

Figure 1 shows the hourly binned average of the EPBs zonal drifts for all observation period as a function of the lunar local

time. The error bars indicate the standard deviation, suggesting a strong variability of calculated EPB zonal drifts. The solid

line represents the least square best fit for 12 hours oscillation, corresponding to the semidiurnal lunar tide period in lunar time.

One can see that the solid line fits very well to the data indicating that the lunar semidiurnal tide was present during the whole

studied period with an amplitude of 3.1 m/s corresponding to 5.6% of the zonal drifts average. Another relevant aspect of being

considered is that, although the amplitude is relatively small, this oscillation is always present in data producing an interesting

day-to-day variability in the dynamics of the EPBs.



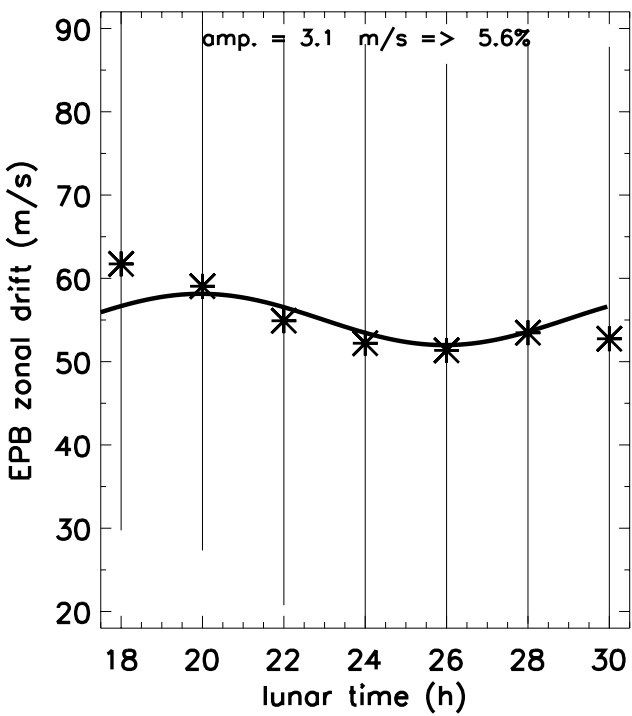

**Figure 1.** EPB zonal drift (stars) obtained from airglow images and semidiurnal lunar tide fit (solid line) as a function of lunar time. Error bars represent the standard deviation at each hourly bin.

Figure 2 shows the EPB zonal drift, according to the seasons, and the semidiurnal lunar tide fits. Panel (a) shows the results for the summer period, considered in this analysis, the months from December to February for all studied years. The calculated $M_2$ amplitude was 7.2 m/s, representing 12.1% of the zonal drift average. Furthermore, there was observed an almost perfect fit to the data, which suggests that the $M_2$ is more pronounced in the variability of the EPB zonal drifts during the summer.

Panel (b) of Figure 2 shows how the $M_2$ acted during the Equinox months (March to May for the Autumns; September to November for the Springs). The amplitude of semidiurnal lunar tide oscillation was 3.0 m/s, which is by about 5.7% of the zonal drift average. The fit was not so good when compared to the Summer period as well.

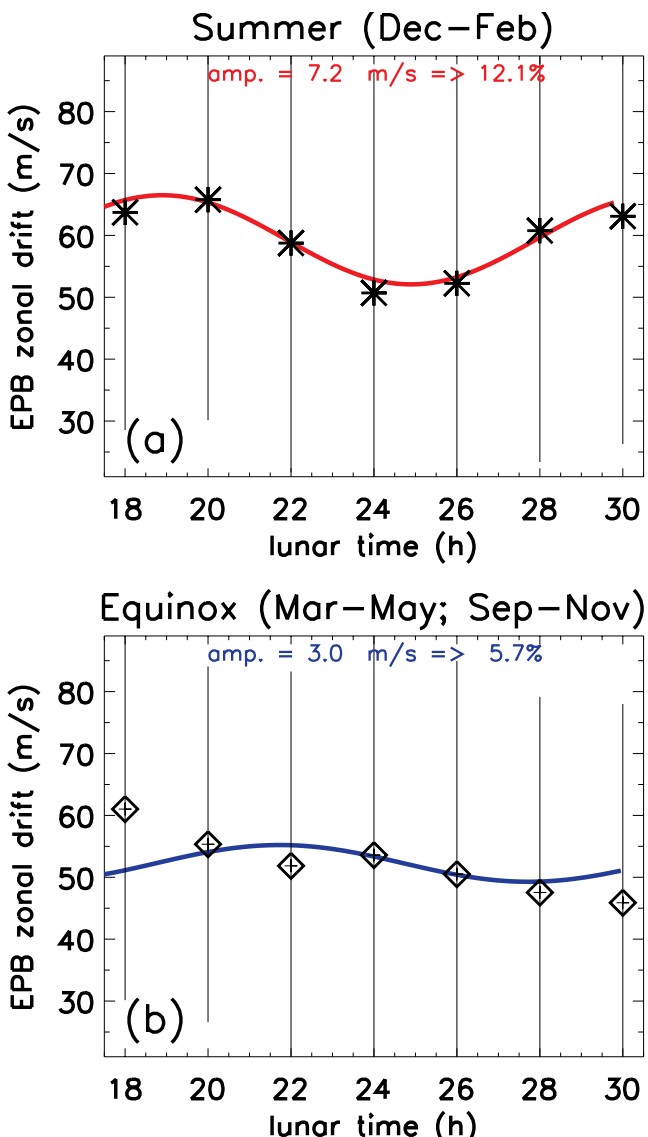

**Figure 2.** EPB zonal drift (stars) for the Summer [stars in Panel (a)] and Equinox [open diamonds in Panel (b)] obtained from airglow images and semidiurnal lunar tide fit [red line in Panel (a) and blue line in Panel (b)] as a function of lunar time. Error bars represent the standard deviation at each hourly bin.

Figures 3 shows the $M_2$ amplitude on EPBs considering the (a) HSA and (b) LSA. The $M_2$ amplitude was over 10% of the zonal drift average for the HSA, i.e., 5.9 m/s. While for the LSA the amplitude was calculated as 1.2 m/s (2.3% of the average). Additionally, the least square best fit was better for the HSA activity than for the LSA, which indicates that the $M_2$ found better conditions to propagate into the thermosphere-ionosphere during the high solar activity.

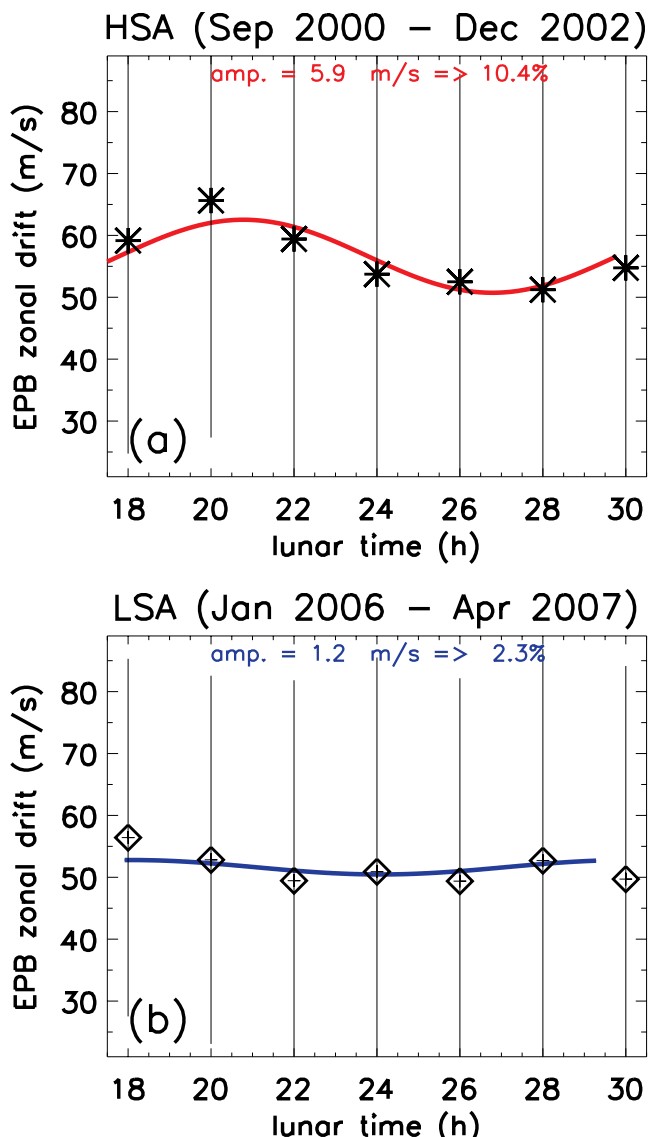

**Figure 3.** EPB zonal drift (stars) for the HSA [stars in Panel (a)] and LSA [open diamonds in Panel (b)] obtained from airglow images and semidiurnal lunar tide fit [red line in Panel (a) and blue line in Panel (b)] as a function of lunar time. Error bars represent the standard deviation at each hourly bin.

## 4 Discussion and summary

The presence of the $M_2$ in the thermosphere-ionosphere system can be understood as a combination of the lunar tide in the geomagnetic field (geomagnetic tide) and neutral wind in the ionosphere (ionospheric tide). The former comes from electrical currents in the E region and the latter is due to the vertical propagation of the tide into the E and F regions. Stening et al. (1999)





showed the importance of this coupling from simulation in a General Circulation Model and Eccles et al. (2011) showed observation results and simulation of the strong connection between the geomagnetic and ionospheric semidiurnal lunar tide. A recent and very comprehensive explanation about the lunar tide in the ionosphere was published by Forbes and Zhang (2019).

During the nighttime near the equator, in general, the F region of ionosphere drifts downward and eastward under magnetic
quiet condition. The zonal drift of the plasma is almost equal to the neutral zonal wind (e.g., Chapagain et al., 2012), i.e., the contribution of the geomagnetic tide seems to be small, primary after midnight.

Based on these aspects, it is expected that the results from Figure 1 represent mostly the contribution of the ionospheric tide. Thus, the value of amplitude of $\sim 3.1$ m/s should be compatible with the amplitude of the lunar tide in the thermospheric zonal wind. Zhang et al. (2014) showed that the $M_2$ presented an amplitude of a few meters per second around the equator
in the zonal wind measured from Gravity Field and Steady-State Ocean Circulation Explorer (GOCE) satellite. Additionally, they showed that the $M_2$ is larger in this region as compared to the geomagnetic contribution from the space perturbations. Additionally, Forbes (1982) extended in Forbes et al. (2014) showed that the semidiurnal tide could propagate directly into the thermosphere, primary the components with long vertical wavelength, which could explain the presence of this oscillation in the thermosphere/EPB zonal drifts.

Regarding the seasonal results retrieved from Figure 2, i.e., during the summer, the $M_2$ amplitude was larger in the EPB zonal drifts than during the equinoxes. The amplitude of the $M_2$ in the zonal wind from the GOCE was large in January and February in the equatorial zone and this behaviour was also predicted by the Global Scale Wave Model Zhang et al. (2014), Additionally, the $M_2$ amplitudes have been studied in the vertical plasma drifts in the equatorial region (e.g. Stening and Fejer, 2001; Fejer and Tracy, 2013) and the results showed similar seasonal dependency, i.e., large amplitudes during the southern
summer.

A full explanation for the seasonal variation in the $M_2$ amplitudes in EPB zonal drift is quite complex due to the complexity of the dynamics involved in the motion of the F region plasma and connections with the E region. Maybe better conditions for propagation of some non-migrating components could the reason for the enhancement of the $M_2$ in zonal and vertical drifts. This hypothesis could be sustained by the observed large $M_2$ amplitudes in the mesosphere and lower thermosphere as in the
temperature (Paulino et al., 2013) as in the zonal wind (Paulino et al., 2015). Furthermore, sudden stratospheric warmings. which are more frequently observed during the southern summer have been pointed out as a mechanism capable of enhancing the amplitude of semidiurnal tides in the equatorial region (e.g., Fejer et al., 2011; Stening, 2011; Forbes and Zhang, 2012; Park et al., 2012; Paulino et al., 2012; Pedatella et al., 2012; Yamazaki et al., 2012; Chau et al., 2015; Maute et al., 2016). During the period of the study of this work, it was observed 4 sudden stratospheric warmings (see Table 2 of Yamazaki, 2014).
Thus, it could also contribute for the $M_2$ amplitudes in the F region drifts during the southern hemisphere summer.

Although in the equatorial region, the neutral winds are less dependent on solar cycle and, consequently, one could expect that driven lunar tide does not present a reasonable solar cycle dependency. The results from Figure 3 showed the $M_2$ amplitudes in the EPB zonal drifts four times larger during the HSA. Forbes and Zhang (2019) showed large $M_2$ amplitudes in the electron concentration during the summer, however, there were slight differences near the equator. Stening et al. (1999) also found large
amplitudes in the vertical drift during he southern hemisphere summer, primary in the nighttime period. In contrast, the vertical





penetration of the semidiurnal lunar tide into the ionosphere from the E region depends on the solar cycle due to the molecular dissipation, which filters some wave components with small vertical wavelengths Forbes (1982); Forbes et al. (2014). These are reasons to believe that the $M_2$ in the zonal drifts can also have an important contribution from the E region as Eccles et al. (2011) showed.

The present work showed that the $M_2$ modulated the EPB zonal drifts in the equatorial region over Brazil from 200 to 2007 and the main results are summarized as following:

   – The amplitude of the semidiurnal lunar tide in the equatorial plasma bubble zonal drifts was 3.1 m/s, which is 5.6% of the mean drifts;

   – It was observed a clear seasonal variability for the $M_2$ amplitudes with high values during the summer of the southern
140       hemisphere when they were compared to the equinox months;

   – The amplitudes of the semidiurnal lunar tide in the equatorial plasma bubble zonal drifts were solar cycle dependent reaching values four times larger during the high solar activity.

An important remark is that further investigations are necessary to understand, primarily, how the zonal drifts of the ionospheric F region respond with large amplitudes during the high solar activity in the equatorial region.

*Data availability.* The used airglow images can be requested to the corresponding author.

*Author contributions.* Igo Paulino wrote the manuscript and made most of the analysis. Ana Roberta Paulino converted solar time to lunar time and revised the manuscript. Amauri F. Medeiros revised the manuscript and helped in the calculation of the EPB zonal drifts. Cristiano M. Wrasse and Hisao Takahashi revised the manuscript. Ricardo A. Buriti revised the manuscript and coordinated the observations.

*Competing interests.* The authors declare that they have no competing interest

*Acknowledgements.* This work has been financed by the Conselho Nacional de Desenvolvimento Científico e Tecnológico (CNPq) under contract 306063/2020-4 and Fundação de Amparo à Pesquisa do Estado da Paraíba (FAPESQ PB) under contract 002/2019. The authors thanks to the Coordenação de Aperfeiçoamento de Pessoal de Nível Superior for making available, even during the pandemic period, the "Periódicos da CAPES", which was fundamental for the construction of this manuscript.




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
