# Peer review of "INFLUENCE OF THE SEMIDIURNAL LUNAR TIDE ON THE EQUATORIAL PLASMA BUBBLE ZONAL DRIFTS OVER BRAZIL"

_Annales Geophysicae, 2021_

## Author Response (AR1)

**Responses to Editor and Reviewers**

**General Comments:**

Dear Dr. Luis Vieira.

We appreciate for considering our manuscript suitable within the scope of Annales Geophysicae. We also thank the three reviewers and Dr. Joseph Olwendo for the comments and suggestions. Out point-by-point responses and the tracked changes manuscript for the Referee #1 can be found as following:

**Reviewer #1:**

REVIEWER:**"Review of 'Influence of the semidiurnal lunar tide on the equatorial plasma bubble zonal drifts over Brazil' by Paulino et al (angeo-2021-38). The paper presents an analysis of OI6300 airglow emissions to determine the semidiurnal lunar tide (M2) contribution to the zonal drifts of equatorial plasma bubbles (EPBs). Analysis of the airglow observations demonstrates that the M2 contributes 5% to the EPB zonal drift variability. The M2 contribution to the EPB zonal drift variability is also found to vary with season and solar cycle. EPBs exhibit significant day-to-day variability, which is important to understand due to their negative impacts on various technologies. The present study is thus a useful contribution to present understanding of the EPB variability, and would be suitable for publication. There are, however, several aspects that I believe need to be clarified prior to publication. More detailed specific comments are provided below. "**

AUTHORS: Thank you for revising our manuscript and for the valuable suggestions that certainly will improve our paper.

REVIEWER:**"1. In lines 56-57, the authors state 'The observations were made between September 2000 and April 2007, centered at new moon periods, resulting in thirteen nights of data per month.' The authors should explain why the observations are restricted to the thirteen days of observations that are centered on the new moon periods. I believe that this is due to the instrument being unable to observe EPBs during the full moon. Restricting the data to new moon periods also limits the lunar local times that can be observed, potentially making the fits to the lunar semidiurnal tide less certain. This limitation should be clearly explained in the text."**

AUTHORS: The reviewer is right. That is a technical limitation because the all sky imager is sensitive to the Moon light. We have just explained it in the manuscript. Regarding the "...making the fits to the lunar semidiurnal tide less certain", the reviewer is right as well, however, choosing the 13 day in a month is enough to cover a full period of the oscillation as can be seen in Figure 1-3. Additionally, the presented results used a long period of observation which made possible a confident statistical analysis. Thank you for the comment and we have also added some words on this point in the manuscript.

REVIEWER:**"2. It is unclear based on the description provided in Section 2 if the analysis places any restriction based on geomagnetic activity. This**

**should be clarified in the text."**

AUTHORS: Thank you for the important comment. We have included the information in the manuscript that there were no magnetic disturbed days from the analysis. The $M_2$ appeared in the EPB zonal drifts independent of the geomagnetic influence, this is relevant for this work. Furthermore, during the high solar activity, when there is more influence of magnetic storms in the ionosphere, the amplitudes of the $M_2$ were higher.

REVIEWER:**"3. Results are presented for Southern Hemisphere summer as well as the equinoxes. Is there a reason why results are not presented for winter? "**

AUTHORS: Thank you for the comment. Yes, the EPBs over Brazil have preferred occurrence from September to March. During the winter, the EPBs develop in a few nights (https://doi.org/10.1016/S1364-6826(02)00089-5), which is not statistically significant to compute the $M_2$. We have also included a statement in the manuscript.

REVIEWER:**"4. One of the conclusions, and intriguing aspects, of the study is the solar cycle dependence, which shows larger M2 amplitudes in the EPB zonal drifts during solar maximum compared to solar minimum. This is opposite what may be expected if the EPB zonal drifts are driven by the in-situ tide that is anticipated to be smaller during solar maximum. I believe that the authors should discuss this result in more detail. In particular, it is important to consider the fact that the analysis was performed for a longer period of solar maximum (September 2000 to December 2002) versus solar minimum (January to April 2007). This has the potential to influence the results and should be clearly discussed. Additional discussions of any previous investigations into the solar cycle variations of the lunar tide in the ionosphere-thermosphere should also be included. "**

AUTHORS: We agree with the Reviewer #1 that it is necessary to expand this discussion. It was the main concern of the other reviewers as well. We have made some comparisons as suggested by the Reviewer #1 and we will revise this topic according to the suggestions of the Reviewers # 2 and # 3 (It will be presented soon). Regarding the second concern, we have used 16 months during the LSA, which we believe to be enough to avoid short term variability in the $M_2$. However, we agree with the reviewer that it is important to mention this difference in the manuscript.

REVIEWER:**"1. Line 6: 'dependents' should be 'dependent'; 2. Line 21: ?motvement" should be 'movement'; 3. Lines 36-37: The sentence beginning with 'As the PRE (vertical motion) ?' is unclear and should be rewritten; 4. Line 130: 'during he' should be 'during the'; Line 135: '200 to 2007' should be '2000 to 2007'. "**

AUTHORS: We appreciate the correction from the reviewer. We have performed all of them in the manuscript according to the suggestions.

**Reviewer #2:**

REVIEWER: "By analyzing the all-sky airglow images taken in Brazil from 2000 to 2007, zonal drift velocity of plasma bubbles are estimated. Based on these velocity data, the authors investigate semidiurnal lunar tide component. This is the topic studied for a long time. This study could provide valuable data. Therefore, this paper is worth publishing in this journal. However, the interpretation and discussion is not enough. Minor revision is needed before its publication. Details are shown below. "

AUTHORS: We really appreciate the relevant contribution suggested by the reviewer.

REVIEWER: "In the discussion section, explanation of definition for geomagnetic tide and ionospheric tide is needed. This reviewer considers that this terminology is not suitable because this reviewer understand as follows. "The geomagnetic tide" is the tide in the E region. The neutral wind variations caused by the tide in the E region generate polarization electric field through the E region dynamo to keep divergence free of the electric current. The polarization electric field generated through the E-region dynamo is transmitted to the F region, causing the ExB drift in the F region. On the other hand, "the ionospheric tide" is the tide in the F region. The neutral wind variation caused by the tide in the F region generates polarization electric field through the F region dynamo. The F-region plasma moves by ExB drift due to the polarization electric field. Therefore, this reviewer considers that "the geomagnetic tide" is the tide in the E region, and that "the ionospheric tide" is the tide in the F region. The authors need to explain the mechanism of the geomagnetic and ionospheric tides. – During daytime, E-region conductivity is higher than the F-region conductivity, so that the polarization electric field through the dynamo mechanism is generated mainly in the E region. However, during nighttime, the plasma density in the E-region disappears due to the recombination. The polarization electric field is mainly generated in the F region and the polarization electric field generated in the E region is negligible. The authors need to argue this point. "

AUTHORS: Thank you for the very didactic explanation. We have revised the discussion considering this suggestion.

REVIEWER: "first line in abstract, "36.5oW": "o" should be a superscript "

AUTHORS: Thank you for the suggestion, we have fixed it.

**Reviewer #3:**

REVIEWER:"**The authors adopted all-sky airglow imager to investigate the zonal drift of EPB and find interesting semidiurnal lunar tide (M2) signatures. The manuscript appears to be a short research letter that only the observational results are provided with inadequate interpretations and discussions.**"

AUTHORS: We thank the reviewer for the comments and suggestions.

REVIEWER:"**The authors' discussion on the solar cycle effects is extremely inadequate, a simple 'must be further investigated' is not an excuse. At least, the author needs to explain why the question cannot be solved in this study? What kind of data might need to resolve the question?**"

AUTHORS: We agree with the reviewer that we have not used the best words in this statement, we have revised it. We are sure of the present results and it is not necessary any further data to conclude on the solar dependence of the $M_2$ in the EPB zonal drifts. What needs to be further studied is the coupling mechanism that is quite complex and it is maybe the "biggest puzzle" for the atmospheric sciences (words from Tsunoda, 2006).

REVIEWER:"**Lines 4-5: Confused with 'the M2 contributes 5.6% to the variability of the EPB zonal drifts'. How the contribution level is determined?**"

AUTHORS: It represents 5.6% of the average EPB zonal drifts. We have clarified it in the manuscript.

REVIEWER:"**Line 33: Be specific about 'nighttime' and 'evening'. Lines 36: Two ?as?.**"

AUTHORS: Thank you for the suggestions, but It is correct. During the first hours the night the zonal drifts are higher.

REVIEWER:"**Line 93: What is the meaning of 'combination' Two independent aspects or the combined two effects?**"

AUTHORS: $M_2$ in the ionosphere responds as the geomagnetic and ionospheric tides. From our point of view, the statement is correct. However, based on the comment of the Reviewer #2, we have improved the discussion on this topic.

REVIEWER:"**Line 106: Rewrite 'Additionally, they showed that the M2 is larger in this region as compared to the geomagnetic contribution from the space perturbations'. Lines 105-107: Two 'Additionally'. Lines 119-120: Rewrite 'as in the temperature (Paulino et al., 2013) as in the zonal wind'. Lines 128-134: Rewrite 'Forbes'. And summary the main idea of this paragraph.**"

AUTHORS: Thank you for the suggestions, we have fixed them.

REVIEWER:"**Lines 128-129: What do mean 'differences near the equator'**"

AUTHORS: We have fixed this statement for a better understanding. Thank you for asking.

**Reviewer #4 (Dr. Joseph Olwendo):**

REVIEWER:"The paper remain significant in understanding the drivers and structuring 0f ionospheric irregularities once initiated. The paper can be accepted for publications but only after a minor revision to the current state. The minor revision is categorized into major corrections and minor corrections. "

AUTHORS: Thanks for the contributions from Dr. Joseph Olwendo, who kindly revised our manuscript.

REVIEWER:"How significant is the 5% value of contribution of M2to the zonal drift velocity? why is the non-negligible. this aspects should be highlighted in the revised manuscript."

AUTHORS: This contribution is relevant because, on average, it is always present with 5% of the EPB zonal drifts. Additionally, $M_2$ is one of the important features for the day-to-day variability of the EPB.

REVIEWER:"what is the scientific explanantion regarding the solar activity and seasonal variations of M2. For example can you expalin why M2 is sttronger during solar max and vice versa."

AUTHORS: It was the most polemic point of this manuscript. However, there are in the literature a couple of works that have pointed out the geomagnetic lunar tide as solar dependent (e.g., Yamazaki and Kosch, 2014). Regarding the ionospheric tide, there are not many reports on it. Assuming that the $M_2$ in the EPB zonal drifts is a combination of these two tides (geomagnetic and ionospheric), we expect that the $M_2$ can be solar dependent as well. Regarding the seasonality, the lunar tide in the MLT is stronger in the December solstice and there were observed enhancement of the $M_2$ associated with SSW events, which are typical from that period of the year.

REVIEWER:"apart from M2 which plays only 5% of the driving forces in the zonal drift, which are the other drivers accountiong for 95% in the zonal drift. "

AUTHORS: The main contribution comes from the solar tide. However, it was observed contributions from other atmospheric waves (gravity and planetary waves, e.g., Abdu et al.,2009, Vadas and Fritts, 2009, Taori et al., 2011, Abdu et al., 2015). There are contributions from the ionosphere-magnetosphere interactions (e.g., Abalde et al., 2009) and we must consider the PRE dynamics (e.g., Kelley and Dao, 2018; Eccles et al., 2015) and the neutral wind daily variation as well (Saito and Maruyama, 2009).

REVIEWER:"Last but not least, the authors should run the revised manuscript in spelling and grammar check before resubmiiting."

AUTHORS: Thank you for the suggestion. We have done it.

REVIEWER:"lines 2-3: "strong day to day ................................near the equator" rewrite the sentence to improve clarity lines 13-14: "they consist.......................................ionosphere. the sentence lacks clarity and must be rewritten. lines 14-15: Changes in lines 13-14 must be matched by a revison in lines 14-15 too for clarity. lines 20-21: PRE in wrongly defined in this section and must be revised. lines 34: the sentence is hanging and is not well connected to the rest. revise this part. The authors should scrutinize the rest of the articles by rrunning the revised version on speclling and grammar check. the above are just afew glaring cases."

AUTHORS: Thank you for the minor revision. We have revised all of them.

**References**

Abalde, J. R., Sahai, Y., Fagundes, P. R., Becker-Guedes, F., Bittencourt, J. A., Pillat, V. G., Lima, W. L. C., Candido, C. M. N., and de Freitas, T. F.: Day-to-day variability in the development of plasma bubbles associated with geomagnetic disturbances, J. Geophys. Res.-Space, 114, A04304, 2009.

Abdu, M. A. and Brum, C. G. M.: Electrodynamics of the vertical coupling processes in the atmosphere-ionosphere system of the low latitude region, Earth Planets Space, 61, 385?395, 2009.

Abdu, M. A., de Souza, J. R., Kherani, E. A., Batista, I. S., MacDougall, J. W., and Sobral, J. H. A. Wave structure and polarization electric field development in the bottomside F layer leading to postsunset equatorial spread F, J. Geophys. Res. Space Physics, 120, 6930- 6940, 2015.

Eccles, J. V., St. Maurice, J. P., and Schunk, R. W.: Mechanisms underlying the pre-reversal enhancement of the vertical plasma drift in the low-latitude ionosphere, J. Geophys. Res.-Space, 120, 4950, 2015.

Kelley, M. C., Dao, E. V. Evidence for gravity wave seeding of convective ionospheric storms possibly initiated by thunderstorms. Journal of Geophysical Research: Space Physics, 123, 4046- 4052, 2018.

Saito, S. and Maruyama, T.: Effects of transequatorial thermospheric wind on plasma bubble occurrences, Journal of the National Institute of Information and Communications Technology, 56, 257-266, 2009.

Sobral, J., Abdu, M., Takahashi, H., Taylor, M., de Paula, E., Zamlutti, C., de Aquino, M., and Borba, G.: Ionospheric plasma bubble climatology over Brazil based on 22 years (1977-1998) of 630nm airglow observations, Journal of Atmospheric and Solar-Terrestrial Physics, 64, 1517-1524, 2002.

Taori, A., Patra, A. K., and Joshi, L. M.: Gravity wave seeding of equatorial plasma bubbles: An investigation with simultaneous F region, E region, and middle atmospheric measurements, J. Geophys. Res.-Space, 116, A05310, 2011

Vadas, S. L., Taylor, M. J., Pautet, P.-D., Stamus, P. A., Fritts, D. C., Liu, H.-L., São Sabbas, F. T., Rampinelli, V. T., Batista, P., and Takahashi, H.: Convection: the likely source of the medium-scale gravity waves observed in the OH airglow layer near Brasilia, Brazil, during the SpreadFEx campaign, Ann. Geophys., 27, 231-259, 2009.

Yamazaki, Y. and Kosch, M. J.: Geomagnetic lunar and solar daily variations during the last 100 years, Journal of Geophysical Research: Space Physics, 119, 6732-6744, 2014.

Tsunoda, R. T. Day-to-day variability in equatorial spread F: Is there some physics missing? Geophys. Res. Lett., 33, L16106, 2006.

---

## Author Response (AR2)

**Responses to Editor and Reviewers**

**General Comments:**

Dear Dr. Luis Vieira.

Thank you for managing this review process. We have tracked the changes in the manuscript and our point-by-point responses can be found as following.

**Referee #2: Dr. Yuichi Otsuka**

REVIEWER: **"The paper has been revised, but still further revision is needed. Especially, in discussion section, explanation in some part is not correct. Also, further detailed discussion is needed. "**

AUTHORS: Thank you for reading and suggesting changes in our manuscript again. Based on your comments, we have revised the manuscript again.

REVIEWER: **"- ll. 17-18: It is better to change 'the EPBs represent plasma irregularities' to 'the EPBs contain plasma irregularities'."**

AUTHORS: Yes, thank you for the suggestion.

REVIEWER: **"- l. 28: It is better to change 'increase its growth rate' to 'initiate the instability'."**

AUTHORS: Thanks for the suggestion. It has accordingly been incorporated.

REVIEWER: **"l. 115, 'The zonal drift of the plasma is almost equal to the neutral zonal wind': The author should describe the reason why the zonal drift of the plasma is almost equal to the neutral zonal wind. The vertical electric field, which induce the zonal plasma drift, is driven through the F region dynamo."**

AUTHORS: We have added the explanation following the reviewer's suggestion. The text was modified to "The zonal drift of the plasma is almost equal to the neutral zonal wind because the vertical electric field, which induce the zonal plasma drifts, is driven through the F region dynamo (e.g., Chapagain et al., 2012). In this case, the contribution of the geomagnetic tide seems to be small, primarily after midnight.".

REVIEWER: **"ll. 131-132, 'A full explanation for the seasonal variation in the M2 amplitudes in EPB zonal drift is quite complex due to the complexity of the dynamics involved in the motion of the F region plasma and connections with the E region.': Further detailed explanation is needed. What is 'connections with the E region'?"**

AUTHORS: We agree with the reviewer that this portion of the manuscript is not clear at all. Thus, we have changed it to "A full explanation for the seasonal variation in the $M_2$ amplitudes in EPB zonal drift is quite complex due to the complexity of the dynamics involved in the motion of the F region plasma, which is strictly related with the E region dynamo, primarily during the daytime.". The meaning of the connection of the E and F region is what we write in this new version. Thank you for the comment.

REVIEWER:"ll. 132-133, 'Maybe better conditions for propagation of some non-migrating components could the reason for the enhancement of the M2 in zonal and vertical drifts.': Further detailed explanation is needed. 'Migrating components' appears suddenly for discussion. It could mean migrating tide. The authors need to explain how the migrating tide is related to the lunar tide."

AUTHORS: Again we have changed the text to be more explicit about the ideas, thank you for the suggestion. We have changed the phase to "The $M_2$ is mainly composed by the migrating component with a secondary contribution of the non-migrating ones, which can play an important role in the global structure of the lunar tide (Paulino et al., 2013). Maybe better conditions for propagation of some tidal non-migrating components could be the reason for the enhancement of the $M_2$ in zonal and vertical drifts". We agree with the reviewer that a confusion could appear relating it with solar tides. However, we believe that now the explanation is consistent. Additionally, we have added the following statement in the Introduction section: "The $M_2$ is composed of migrating, which follow the motion of the Moon, and non-migrating components, which can propagate to the east or west."

REVIEWER:"ll. 146-152: The authors argue contribution of the E region on the lunar tide. During nighttime, the E region plasma disappears due to the rapid recombination. Consequently, the E region does not affect the lunar tide in the F region, observed in this study. On the other hand, equatorial electro jet can be measured only during daytime. The current result is compared with other papers that study the tide during daytime by analyzing the magnetic field data. The authors need to consider the difference between daytime and nighttime. "

AUTHORS: Thank you for the important comment. We are aware that our data is poorly affected by the E region dynamo. We included the comparison with the E region works in order to show that the lunar tide has solar cycle dependency in other ionospheric parameters. We have revised the paragraph to avoid a misunderstanding in this meaning. The revised version is "It is expected that our results have weak influence of the E region dynamo because during the nighttime the recombination acts quickly to suppress the E region. Complementarily, it is established that the vertical penetration of the semidiurnal lunar tide into the ionosphere from the E region depends on the solar cycle due to the molecular viscosity, which filters some wave components with small vertical wavelengths (Forbes, 1982; Forbes et al., 2014). Yamazaki and Kosch (2014) showed a clear solar dependence of the geomagnetic lunar tide over the last century and other works have also showed solar dependence of the lunar tide modulation in the Equatorial Electro Jet (e.g., Eccleset al., 2011; Luhr et al., 2012; Yizengaw and Carter, 2017). Additionally, Eccles et al. (2011) showed $M_2$ oscillation modulation parameters of the F region as well. Those results contribute to understanding that the lunar tide is solar dependent in some ionospheric parameters. Thus, these are reasons to believe that the $M_2$ in the zonal drifts can also be solar dependent. "

REVIEWER:"l. 147, 'due to the molecular dissipation': How is the molecular dissipation related to this argument?"

AUTHORS: We have changed the statement to '...depends on the solar cycle due to

the molecular viscosity, which filters some wave components with small vertical wavelengths...'. We think it is enough to address this point.

REVIEWER: **"ll. 153, 'parameters of F as well': What is 'F'?"**

AUTHORS: We missed the word region after 'F'. We have fixed it.

**Reviewer #3: Dr. Xin Wan**

REVIEWER:"**I'm satisfied with the authors' corresponding regards on my major concerns. However, I would suggest the authors to present specific actions they made in the response letter, regarding the reviewers' minor comments, rather than posting that 'We have clarified it in the manuscript' or 'we have fixed them'.**"

AUTHORS: We thank Dr. Wan for revising our manuscript once more. The reviewer is right. Our apologies for that. We tried to shorten the response, but we agree that some words are necessary to explain what was done and it can enhance the debate. We will try to avoid these simple responses next time.

REVIEWER:"**Lines 34-35: present specific information on the 'seasonal and solar activity dependencies' A few typo issues still exist. **"

AUTHORS: Unfortunately, we could not see the typo in these lines. We suppose the typo is in the following line. Additionally, we have revised the whole manuscript for typo as you can see in the tracked changes file.

REVIEWER:"**Line 36: 'disturb' should be 'be totally disturbed'.**"

AUTHORS: The reviewer is right. We have corrected it according to the suggestion.

REVIEWER:"**Line 153: 'F' should be 'F region'.**"

AUTHORS: Yes, the reviewer is correct. You have changed it.

---

## Author Response (AR3)

**Responses to Editor**

Dear Dr. Luis Vieira.

Thank you for revising our manuscript and for the important suggestion: "Thank you very much for revising the manuscript taking into account the reviewer's suggestions. I recommend publishing subject to minor revision in Figures 1-3, which consists of including the uncertainties for your estimates of the quantities."

We have included the uncertainties for each estimated amplitude on the top of the panels and we have also included this information in the captions.

Best regards,

The authors